# Explore the shared molecular mechanism between dermatomyositis and nasopharyngeal cancer by bioinformatic analysis

Xiuqin Zhong[1‡], Jingjing Shang[1‡], Rongwei Zhang[1], Xiuling Zhang[1], Le Yu[1], Haitao Niu[2]*, Xinwang Duan[1]*

1 Department of rheumatology and immunology, The Second Affiliated Hospital, Jiangxi Medical College, Nanchang University, Nanchang, China, 2 School of Medicine, Key Laboratory of Viral Pathogenesis & Infection Prevention and Control, Guangzhou Key Laboratory of Germ-free Animals and Microbiota Application, Jinan University, Guangzhou, China

‡ XZ and JS are contributed equally to this work and co-author.
* dxw_efyfsmyk@163.com (XD); htniu@jnu.edu.cn (HN)

## Abstract

### Background

Dermatomyositis (DM) is prone to nasopharyngeal carcinoma (NPC), but the mechanism is unclear. This study aimed to explore the potential pathogenesis of DM and NPC.

### Methods

The datasets GSE46239, GSE142807, GSE12452, and GSE53819 were downloaded from the GEO dataset. The disease co-expression module was obtained by R-package WGCNA. We built PPI networks for the key modules. ClueGO was used to analyze functional enrichment for the key modules. DEG analysis was performed with the R-package "limma". R-package "pROC" was applied to assess the diagnostic performance of hub genes. MiRNA-mRNA networks were constructed using MiRTarBase and miRWalk databases.

### Results

The key modules that positively correlated with NPC and DM were found. Its intersecting genes were enriched in the negative regulation of viral gene replication pathway. Similarly, overlapping down-regulated DEGs in DM and NPC were also enriched in negatively regulated viral gene replication. Finally, we identified 10 hub genes that primarily regulate viral biological processes and type I interferon responses. Four key genes (GBP1, IFIH1, IFIT3, BST2) showed strong diagnostic performance, with AUC>0.8. In both DM and NPC, the expression of key genes was correlated with macrophage infiltration level. Based on hub genes' miRNA-mRNA network, hsa-miR-146a plays a vital role in DM-associated NPC.

**Data Availability Statement:** All relevant data are within the paper and its Supporting Information files.

**Funding:** This study was supported by a Key Research and Development Program of Jiangxi Province grant awarded to XD [20192BBGL70024].

**Competing interests:** The authors have declared that no competing interests exist.

## Conclusions

Our research discovered pivot genes between DM and NPC. Viral gene replication and response to type I interferon may be the crucial bridge between DM and NPC. By regulating hub genes, MiR-146a will provide new strategies for diagnosis and treatment in DM complicated by NPC patients. For individuals with persistent viral replication in DM, screening for nasopharyngeal cancer is necessary.

## Introduction

Dermatomyositis (DM) is a specific inflammatory myopathy with an unclear pathophysiology. Genetic, immunological, and environmental factors contribute to DM development. HLA-DR haploids, MHC polymorphisms, DNA methylation, and non-encoding RNA play important roles in DM [1]. Infections, drugs, ultraviolet radiation, and smoking can trigger autoimmune response. UV irradiation induces DM-related antibodies, like anti-mi2 and anti-tif1. Immune tolerance is disrupted by viral infections. Complement activation causes capillary destruction, ischemic infarction and even muscle atrophy. In addition, type I interferon signaling and myositis antibodies have a major role in the development of DM [2]. In subjects with a background genetic susceptibility to dermatomyositis, activation of MDA5 during viral infection may lead to overexpression of type I IFN by p DCs, resulting in enhanced antigen presentation by antigen-presenting cells (APCs) and antibody production by plasma cells. type I IFN stimulates the maturation of B cells into plasma cells, which in turn produces anti-MDA5 autoantibodies. Anti-MDA5 autoantibodies are involved in the formation of immune complexes (ICs), which promote IFN production by stimulating Toll-like receptor 7 (TLR7). The interaction between anti-MDA5 autoantibodies and type I IFN creates a vicious circle [3]. In addition, type I IFN can lead to myotube mitochondrial damage by increasing reactive oxygen species production [4].

Multiple malignancies are predisposed to DM patients. Chinese studies have found that DM has a high risk of complications caused by nasopharyngeal carcinomas (NPC) [5]. In Taiwan, NPC is the most common (22.95%) in DM patients, followed by lung, breast, and ovarian cancers [6]. High-risk factors for DM complicated by NPC include males, age > 40 years, severe rash or muscle damage, dysphagia, and muscle weakness. NPC typically occurs 1 year after the diagnosis of DM [7]. There is poor prognosis for dermatomyositis complicating nasopharyngeal carcinoma patients. Stronger drug resistance and more severe radiotherapy complications lead to poor survival in patients [7, 8]. Tumor radiotherapy enhances the anti-tumor immune response of CD8+ T-cells by radiation-induced activation of STING signaling and subsequent type I interferon (IFN-I) production [9]. Excessive type I IFN production leads to skin and muscle damage, thereby exacerbating dermatomyositis symptoms. Various chemotherapeutic agents such as cisplatin or STING agonists induce IFN-I production to exert anti-tumor effects. However, tumor cells may inhibit STING levels and IFN-I production by increasing the expression of tryptophan metabolites or high expression of the aromatic hydrocarbon receptor AhR, enhancing patient resistance [10].

However, the pathogenesis of DM complicates NPC remains unclear. Our study provided a new perspective on exploring the mechanisms of DM comorbid with NPC. This study will provide new strategies for diagnosis and treatment in DM patients, and contribute to improve prognosis.

## Methods

### GEO dataset download and process

From the NCBI GEO database (Home - GEO - NCBI (nih.gov)), we searched for DM and NPC gene expression profiles based on the following criteria: i) there must be more than 20 samples; ii) each dataset must include cases and healthy controls; iii) skin biopsy specimens (DM group) or nasal tissues (NPC group) should be used for gene expression analysis. Finally, four datasets (GSE46239, GSE12452, GSE142807, GSE53819) were downloaded (**S2–S5 Tables**). The datasets GSE46239 and GSE53819 were selected as training sets. GSE142807 and GSE53819 were used as validation sets (**S1 Table**). After the data matrix was transformed by Log2, the probe was annotated with gene names according to the platform annotation file.

### Co-expression network construction and identification of key gene modules

Using R-package WGCNA [11], we identified gene sets significantly associated with DM and NPC phenotypes, respectively. More than 20,000 genes have been identified in datasets GSE46239 and GSE12452, and removed genes that are not significantly differentially expressed across samples (based on standard deviation≤0.5). Using the function "sft$powerEstimate" to set the appropriate soft power value, the GSE46239 dataset applied a β value of 9 and the dataset GES12452 had a β value of 5. First, the Adjacency matrix was built according to β, then the topological overlap matrix (TOM) and the corresponding dissimilarity(1-TOM) were transformed from the adjacency matrix. By hierarchical clustering, gene trees were further constructed based on the corresponding dissimilarity. Each branch of the tree corresponds to a distinct gene module. Gene modules with similar expression patterns were grouped into one module eigengene (ME) with the following parameters: "minModuleSize = 50", "deepSplit = 2" and "cutHeight = 0.25". A Pearson correlation test was used in WCGNA to calculate the correlation between MEs and clinical traits.

### Identified subnetwork and shared key genes of DM and NPC

Using the Venn online tool (Draw Venn Diagram (ugent.be)), we calculated the share genes between DM and NPC. Gene Ontology (GO) and Kyoto Encyclopedia analyses (KEGG) were performed through the DAVID website (DAVID Functional Annotation Bioinformatics Microarray Analysis (ncifcrf.gov)) [12], P-value <0.05 was considered significant. Using string tools (STRING: functional protein association networks (string-db.org)) [13], protein-protein interaction networks (PPIs) were analyzed. The PPI subnetwork was subsequently constructed using the MCODE k-means algorithm with parameters "degree cutoff = 6", "node score cutoff = 0.2", and "K-core = 6". A functional analysis of subnetworks was performed using the ClueGO plug-in in Cytoscape (version 3.9.1) [14].

### Validation of the hub genes by DEGs analysis

By using R package "limma" we acquired DEGs between case and control groups. Genes in datasets GSE142807 and GSE53819 have been categorized as up-regulated or down-regulated groups based on the following cutoff value: "|logFC| = 1", " adjusted p-values<0.05". In Venn, common DEGs between DM and NPC were identified, and ClueGO identified biological processes.

### Interaction of genes and functional analysis

GeneMANIA (GeneMANIA) aggregates hundreds of datasets and millions of interactions from GEO, BioGRID, IRefIndex and I2D [15]. Using GeneMANIA, we validated genes that could potentially share functions with the hub genes.

### Construction of miRNA-mRNA network of hub genes

We used miRTarBase (miRTarBase: the experimentally validated microRNA-target interactions database (cuhk.edu.cn)), a database containing over 360,000 miRNA-target interactions and validated them by reporter assays, Western blots, CLIP-Seq and microarrays [16] to identify miRNAs notably associated with key genes. MiRNAs associated with DM and NPC were collected from miRWalk. Version 3 of miRWalk (Home - miRWalk (uni-heidelberg.de)) stores predicted data obtained with a machine learning algorithm, including experimentally verified miRNA-target interactions [17]. The miRNA-mRNA network was created with Cytoscape software.

### Correlation analysis between immune cell infiltration and hub genes

CIBERSORT estimates immune cell infiltration by gene expression data. The immune cell infiltration matrix was downloaded from the CIBERSORT website (CIBERSORTx (stanford.edu))[18]. Using the CIBERSORT-R package, we predicted the level of infiltration of 22 different immune cells within pathological tissues, with P-value < 0.05. For assessing hub genes' significance for immune cell infiltration, spearman correlation analysis was conducted.

### Diagnostic performance and gene set enrichment analysis of hub genes

The pROC analysis in R-package was used to assess the ability of pivotal genes to predict disease. In order to assess pathways and molecular mechanisms between genes with differing expression levels, GSEA software (version 4.3.2) was used [19]. The pivotal genes were divided into two groups based on their expression levels. Nominal P-value< 0.05 and |normalized enrichment scores (NES) | > 1 are considered significant. GSEA enrichment results were visualized with R-ggplot2.

### Statistical analysis

R software (version 4.0.0) was applied for statistical analyses. P-value<0.05 was considered statistically significant. **Fig 1** shows the flow of the study.

## Results

### Training set: Co-expression modules of DM and NPC

Gene co-expression dendrograms and module traits were constructed using the R-WGCNA package. Of the seven co-expression modules in GSE46239, only the red module (r = 0.38, p = 0.005) showed significantly positive effects on DM. It included 730 genes in total. As shown in **Fig 2C and 2D**, GSE12452 was clustered into 9 different co-expression modules, of which 4 gene modules (blue: r = 0.74, p = 4e-08, brown: r = 0.51, p = 6e-04, magenta: r = 0.47, p = 0.002, purple: r = 0.45, p = 0.003) showed a significant positive correlation with NPC, which included 684, 749, 70, and 64 genes, respectively. The turquoise module (r = -0.85, $p$ = 3e-12), green module (r = -0.85, $p$ = 2e-12), and pink module (r = -0.57, $p$ = 1e-04) were negatively correlated with NPC. Further analysis will focus on the key gene modules that positively relate to the clinical trait.

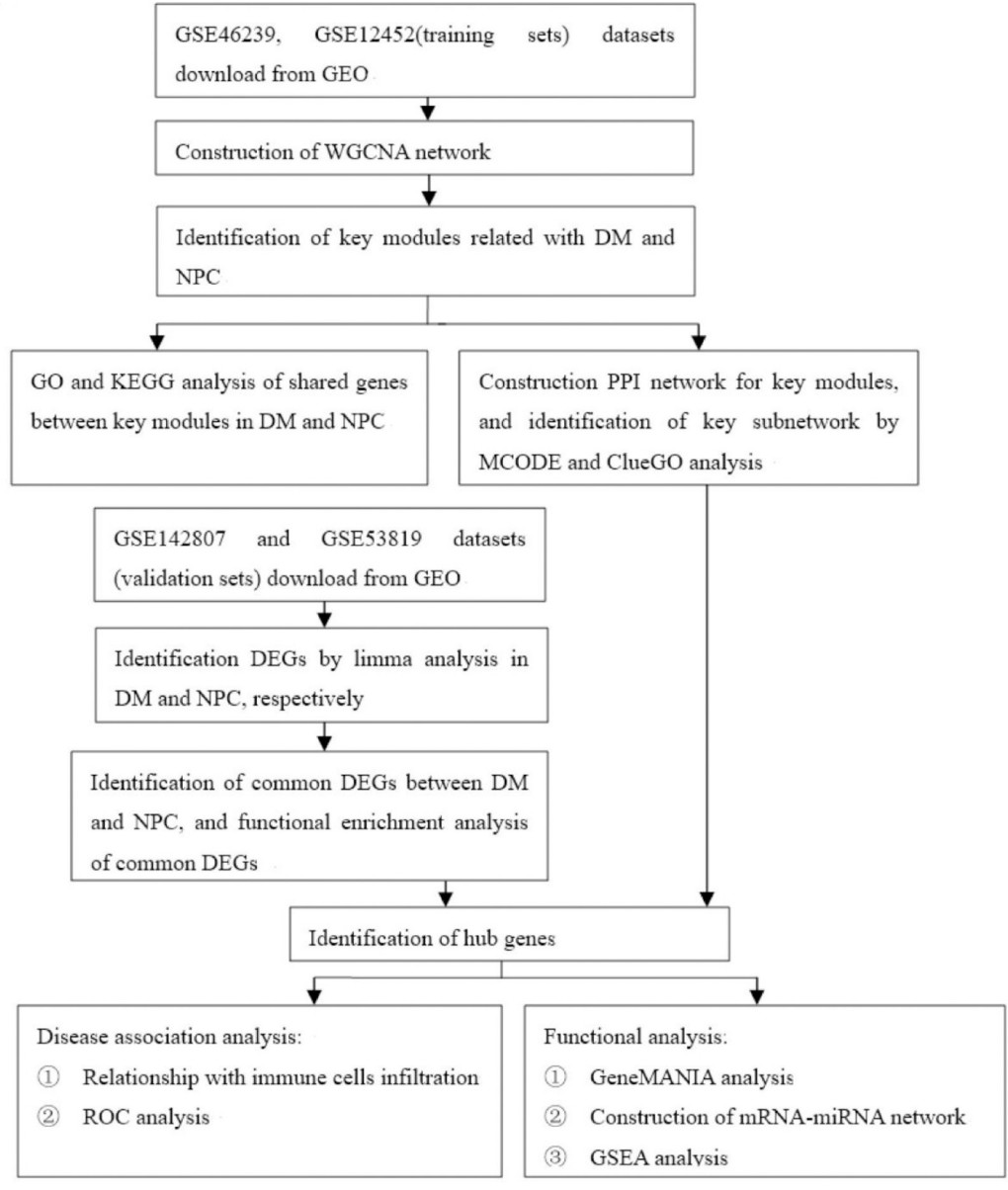

**Fig 1. Research flow diagram.**

### Functional enrichment analysis identified the share genes of DM and NPC

DM and NPC shared 189 genes between their positive correction modules, as illustrated in
**Fig 3A**. In DAVID, GO and KEGG enrichment analysis were conducted for these genes.
According to GO analysis, the first three significant biological processes included defense
response to virus, response to virus, and negative regulation of viral genome replication
(**Fig 3C**). Immune-related pathways, such as innate and adaptive immune responses, also
played an important role. KEGG pathways enrichment analysis identified viral protein interaction with cytokines and cytokine receptors, Toll-like receptor signaling pathway, and Coronavirus disease-COVID-19, which were all involved in viral infection (**Fig 3B**). PPI networks
were built for the key modules of DM and NPC, respectively. The MCODE algorithm

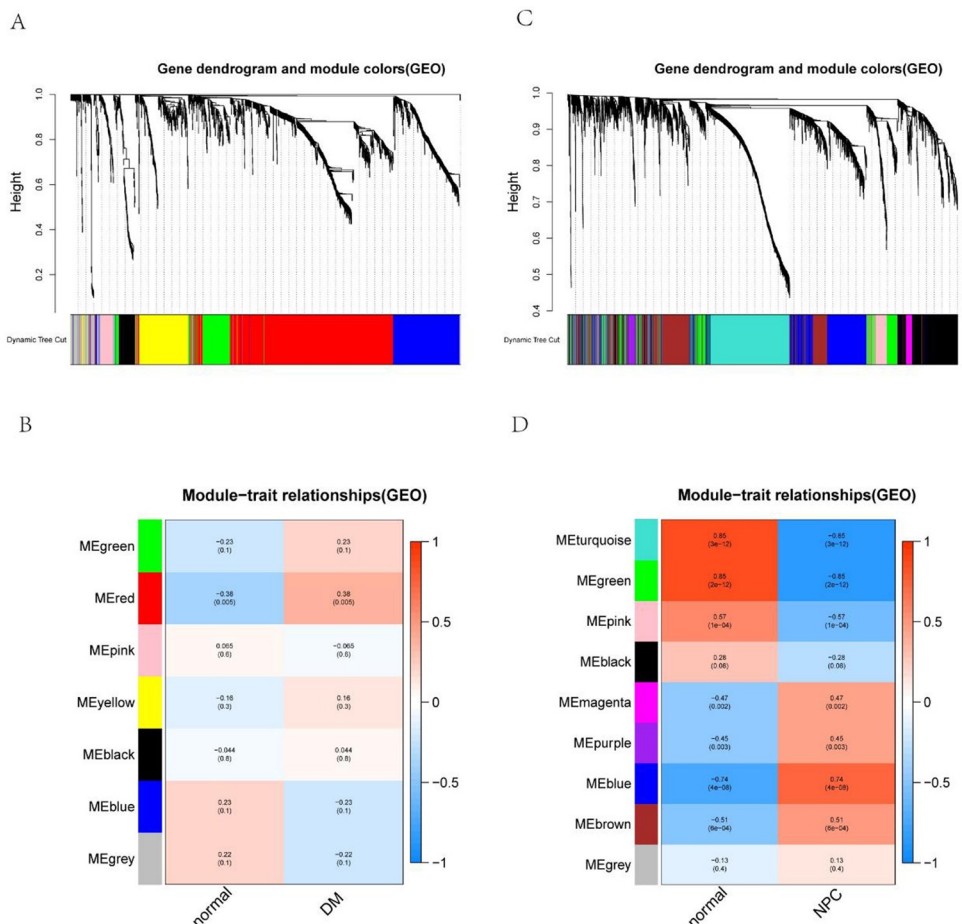

**Fig 2. Weighted gene co-expression network analysis (WGCNA).** (A) The cluster dendrogram of co-expression genes in DM. (B) Module-trait relationships in DM. Per cell includes the corresponding correlation and *p*-value. (C) The cluster dendrogram of co-expression genes in NPC. (D) Module-trait relationships in NPC. Per cell includes the corresponding correlation and *p*-value. DM, dermatomyositis; NPC, nasopharyngeal carcinoma.

clustered the PPI network (**S1A Fig in S1 File**) of the red module in DM into 3 subnetworks (**Fig 4A–4C**), with cluster 1 having 33 nodes and 526 edges (score = 32.875). PPI network (**S1B Fig in S1 File**) of purple module in NPC was clustered into 1 subnetwork (**Fig 4D**) with 29 nodes and 354 edges (score = 25.286). Additionally, brown and blue modules of NPC formed 1 and 3 subnetworks independently (**S2A-S2F Fig in S1 File**). The shared pathways between DM and NPC are that negatively regulation of viral genome replication and response to type I interferon, as visualized by ClueGO plug-in (**Fig 4E and 4F**). Finally, 49 shared genes were obtained from the red modules in DM and purple module in NPC, defined as gene set 1.

## Validation set: Identification of hub genes through DEGs analysis

The volcano image of DEGs in DM and NPC datasets were generated using R-package limma (**Fig 5A and 5B**). There are 5587 DEGs in GSE142807, with 2626 up-regulated genes and 2961 down-regulated genes. Among the 2078 DEGs in GSE53819, 864 genes were down-regulated and 1214 genes were up-regulated. There are 52 common down-regulated DEGs between DM and NPC (**Fig 5C**), named gene set 2. In ClueGO functional enrichment analysis (**Fig 5D**), gene set 2 primarily enriched in negative regulation of viral genome replication, similar to

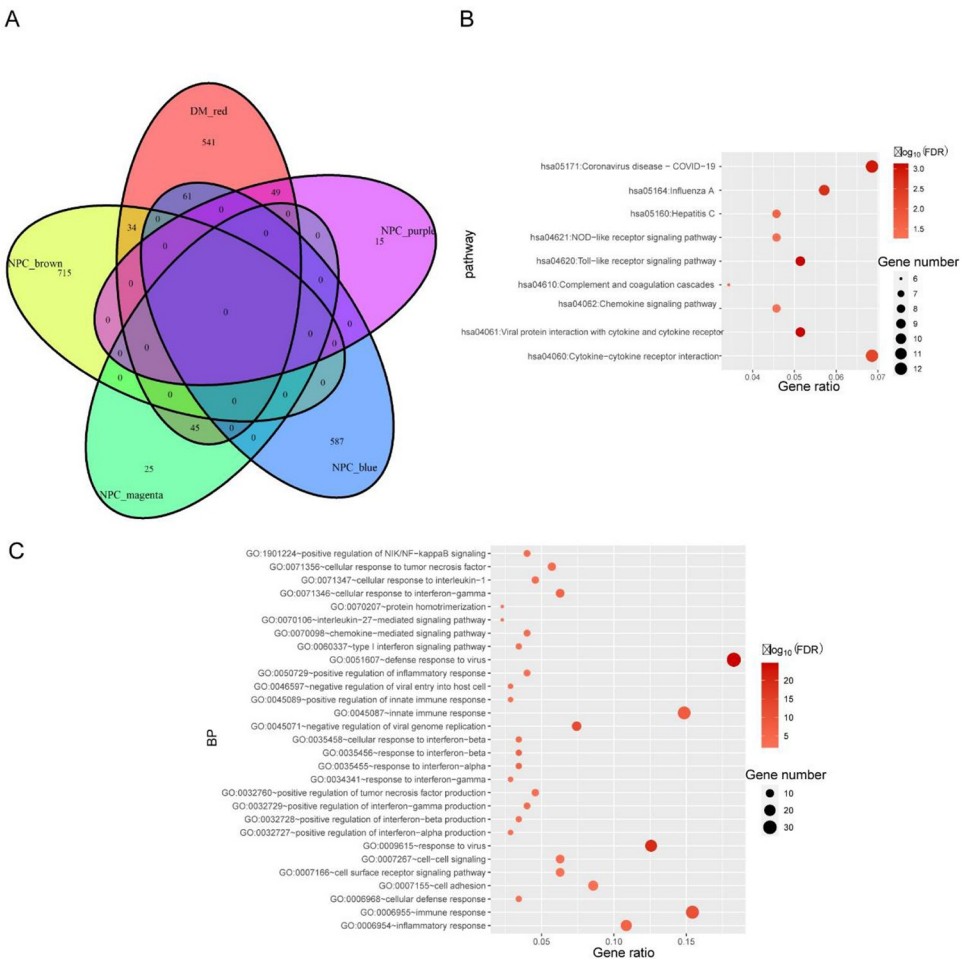

**Fig 3. Function Enrichment analysis results of the shared genes.** (A)The shared genes between the red modules of DM and the blue module, brown module, purple module and magenta module of NPC. (B, C) The enrichment analysis results of GO and KEGG pathway. The size of the circle represents the number of genes involved, and the abscissa represents the frequency of the genes involved in the term total genes. Adjusted P-value<0.05 was considered significant. GO, gene ontology; KEGG, kyoto encyclopedia of genes and genomes.

gene set 1. Eventually, 10 intersection genes were identified between gene set 1 and 2 (**Fig 6A**) and selected as hub genes. There were 92 common up-regulated genes between DM and NPC**, and the** enrichment analysis results were shown in **S3A, S3B Fig in S1 File**.

## PPI and miRNA-mRNA network construction of hub genes

GeneMANIA prediction algorithm was used to analyze co-expression, physical interactions, colocalization, and pathway analysis of 20 interacting genes with hub genes and their PPI network. It was also discovered that virus-associated pathways have also been exploited (**Fig 6B**). Our study also identified miRNAs associated with hub genes that have a negative regulatory effect on gene expression. Based on the miRWalk database, 2114 miRNAs related to DM and 2221 miRNAs related to NPC were predicted (**S6, S7 Tables**). According to the miRTarbase, 58 different miRNAs target the hub genes (**S8 Table**). Finally, we discovered 49 shared miRNAs that targeted hub genes in both DM and NPC. The miRNA-mRNA network was visualized by Cytoscape software, including 57 nodes (8 for mRNA, 49 for miRNA) and 56 edges

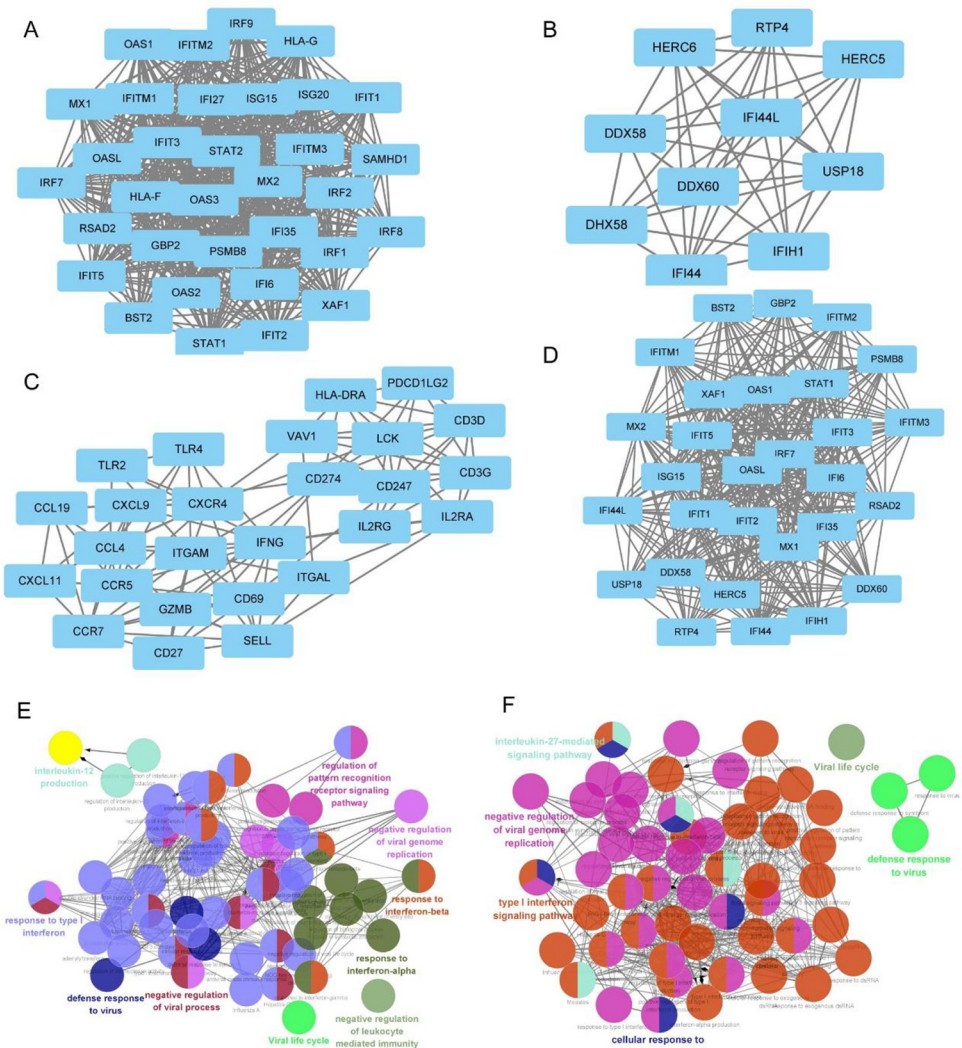

**Fig 4. The PPI network and clusters analysis.** (A) The subnetwork 1 was withdrawn from the red module of DM. (B) The subnetwork was withdrawn from the purple module of NPC. (C, D) The proportion of each GO terms group in the total of subnetworks in DM and NPC, respectively. (E, F) The interaction network of GO terms and KEGG pathways generated by the Cytoscape plug-in ClueGO of the red module in DM and the purple module in NPC, respectively.

(**Fig 7**). Has-miR-146a-5p regulates the majority of hub genes (*IFI44L*, *ISG15*, *IFIT1*, *IFI44*, *and IFIT3*). Downregulation of antiviral genes by hsa-miR-146a-5p could play a role in pathogenesis of DM complicated by NPC.

## Diagnosis performance and immune-associated analysis of hub genes

We used the CIBERSORT algorithm and spearman analysis to confirm the relationship between immune infiltration and hub genes. Dendritic cells resting and Tregs were negatively correlated with hub genes in DM. In addition, macrophage M2 and T cell memory resting were negatively related to most hub genes in NPC. DM and NPC all showed a strong correlation between macrophage M1 and hub genes (**Fig 8A–8D**). Hence, down-regulation of hub genes can inhibit the inflammatory response and immune checkpoint of macrophage M1, but

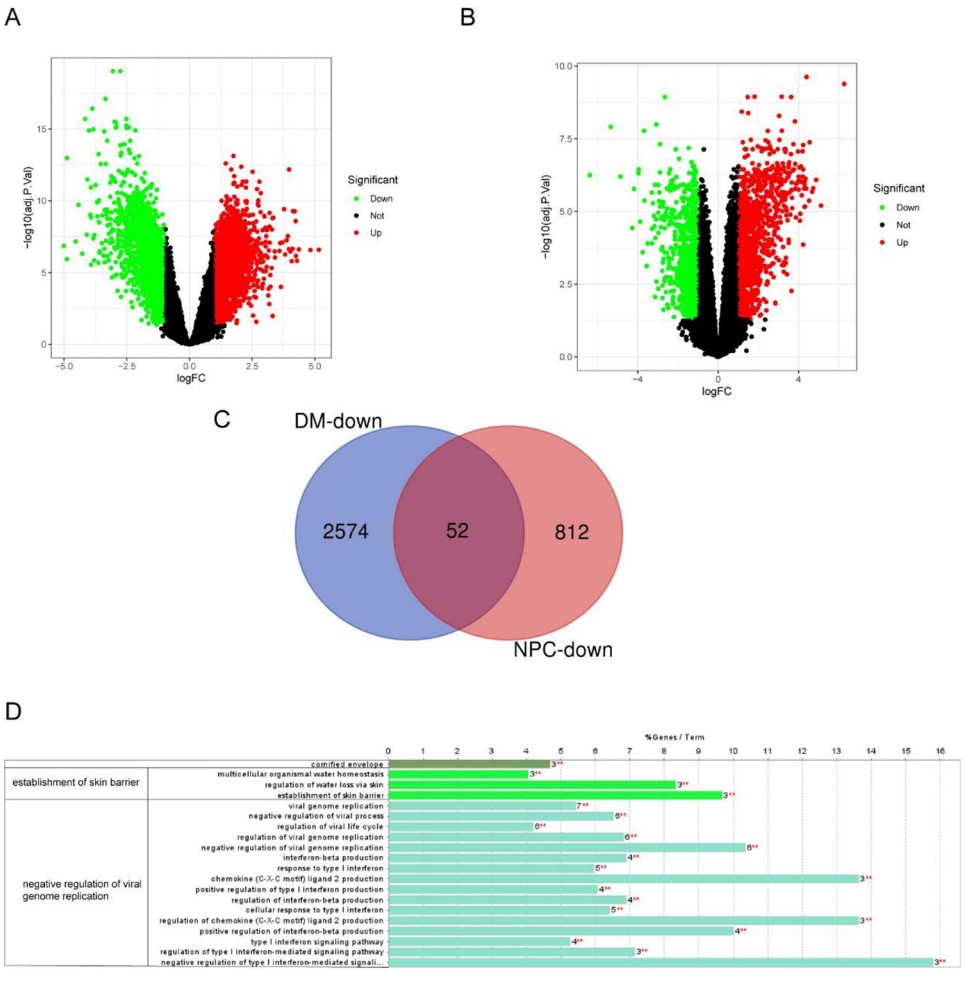

**Fig 5. Volcano image and ClueGO enrichment analysis of common DEGs.** (A, B) The volcano map of GSE142807 and GSE53819. Upregulated genes are marked in light red; downregulated genes are marked in light green. (C) The two datasets showed an overlap of 52 the common downregulated DEGs. (D) The GO biological process analysis of the common DEGs. one or two key words were used to summarize their main biological functions. DEGs; differentially expressed genes.

macrophage M2 may assist the immune escape of tumor cells, thus promoting tumor growth [20]. In addition, the down-regulation of the central gene inhibits the immune inflammatory response and increases the chance of viral infection. Viral infection is not only involved in the occurrence of autoimmune diseases, but also closely related to the occurrence and development of NPC [21]. Using the R-pROC package, hub genes performed well in diagnostic performance. DM and NPC diagnosis may be assisted by *BST2*, *IFIT3*, *IFIH1* and *GBP1*, whose AUC values were greater than 0.8 (**Fig 9A–9D**).

## GSEA result of hub genes

To further identify the role of hub genes on macrophage and viral processes, genes with higher disease evaluation performance (AUC value>0.80) were selected. GSEA was applied to assess the diverse pathway in higher and lower expression groups of hub genes in DM and NPC. Positive regulation of macrophage cytokine production, macrophage differentiation, and

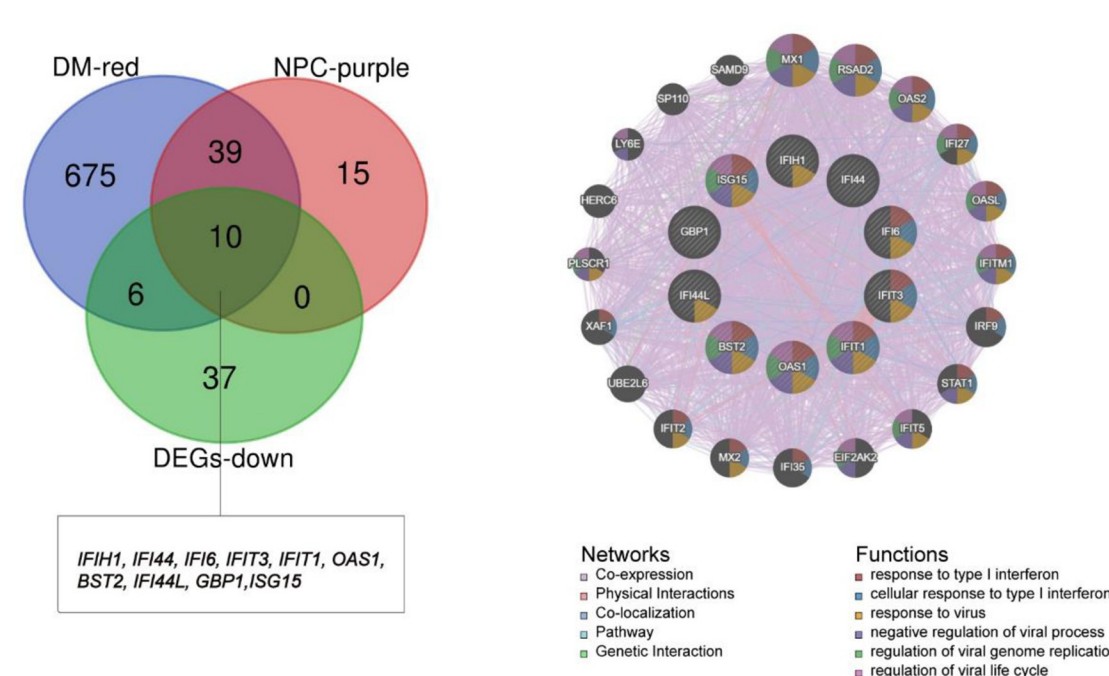

**Fig 6. Venn diagram and co-expression network of hub genes.** (A) The Venn diagram showed that 10 overlapping hub genes have screened out from the key modules in disease and common DEGs. (B) Hub genes and their co-expression genes were analyzed via GeneMANIA.

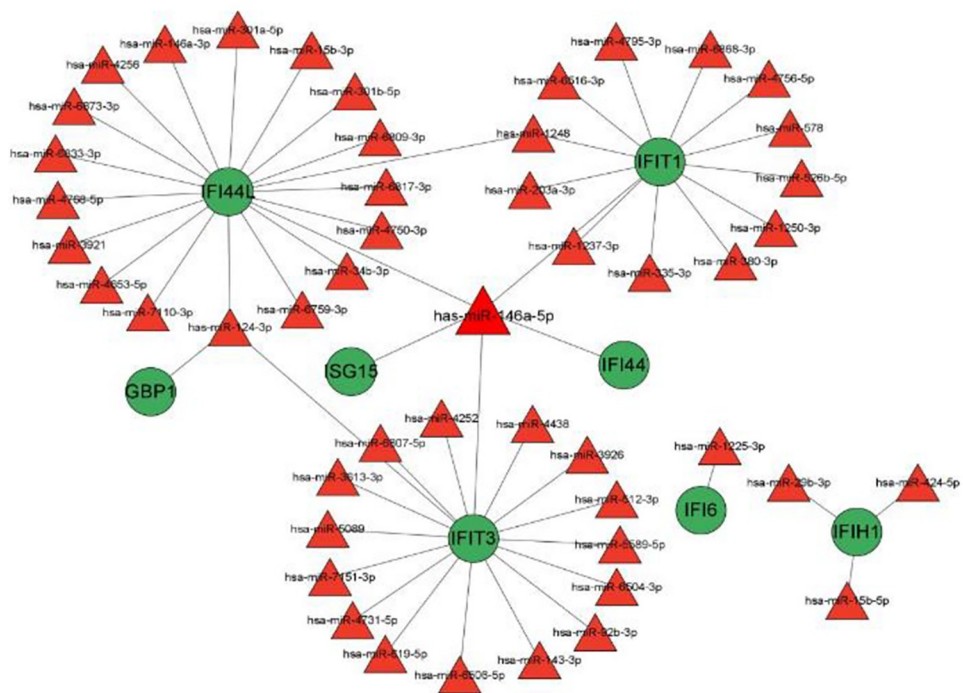

**Fig 7. Genes–shared miRNAs regulatory network.** Red triangles represent miRNAs and green circles represent hub genes.

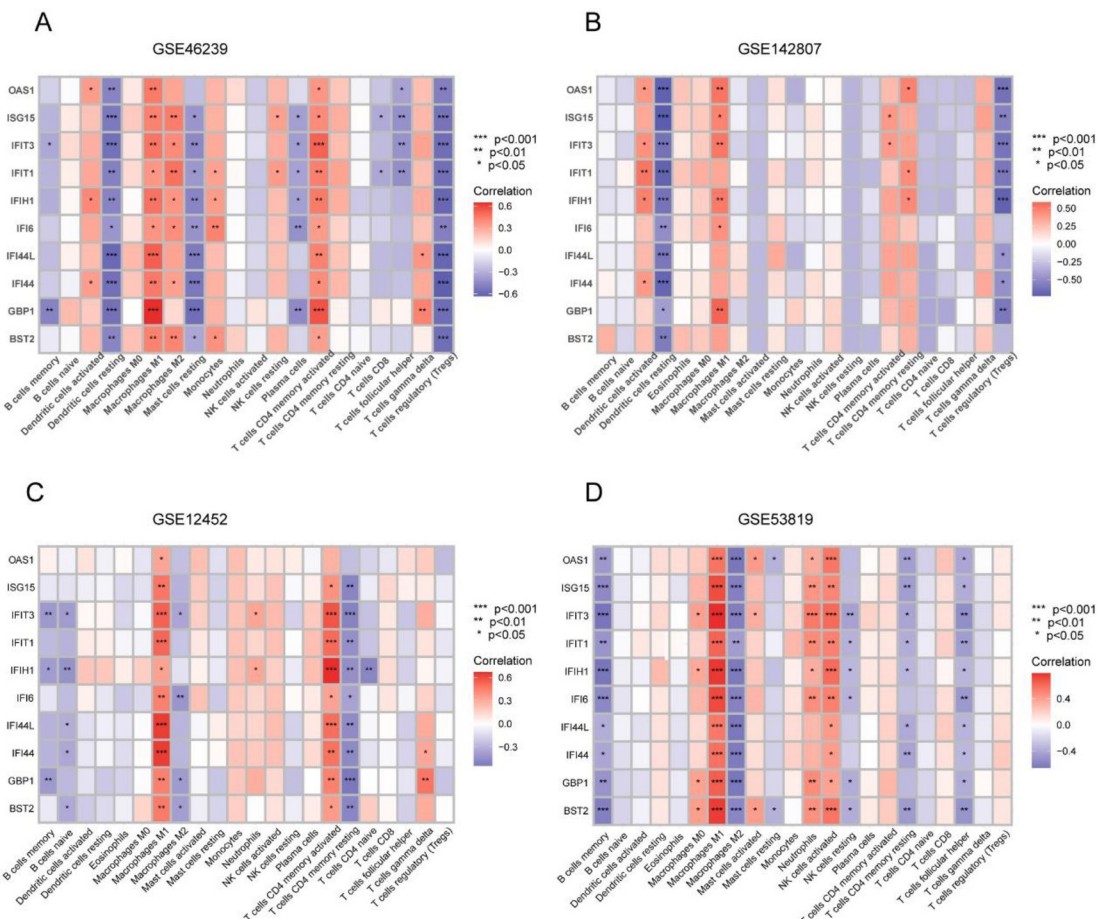

**Fig 8. The relationship between pivot genes and immune cells infiltration.** (A-D) The heatmap of the correlation between the 22 immune cells and hub genes in the four datasets, respectively. Colored squares represent the strength of the correlation; the red color represents a positive correlation, and the blue color represents a negative correlation. The deeper the color, the stronger the correlation. ***, P-value <0.001; **, P-value<0.01; *, P-value<0.05.

regulation of viral genome replication were the common pathway with the higher expression of *IFIH1*, *GBP1and BST2* in GSE53819 and GSE142807(**Fig 10A–10F**):

'GOBP_NEGULATION_OF_VIRAL_GENOME REPLICATION',
'GOBP_POSITIVE_REGULATION_OF_MACROPHAGE_CYTOKINE_PRODUCTION',
'GOBP_REGULATION_OF_VIRAL_PROCESS',
'GOBP_MACROPHAGE_CYTOKINE_PRODUTION',
'GOBP_REGULATION_OF TYPE_ I_ INTERFERON_ MEDIATED_ SIGNALING PATHWAY'.

Regulation viral process and macrophage cytokine production were closely related with the high expression of *IFIH1*, *GBP1and BST2*.

## Discussion

### Viral genome replication in DM and NPC

The viral infection often triggers connective tissue diseases like systemic lupus erythematosus (SLE) and dermatomyositis. Coxsackievirus RNA was detected in muscle biopsies of individuals with DM, along with HIV, Hepatitis B, and Influenza A viruses, all associated with DM onset

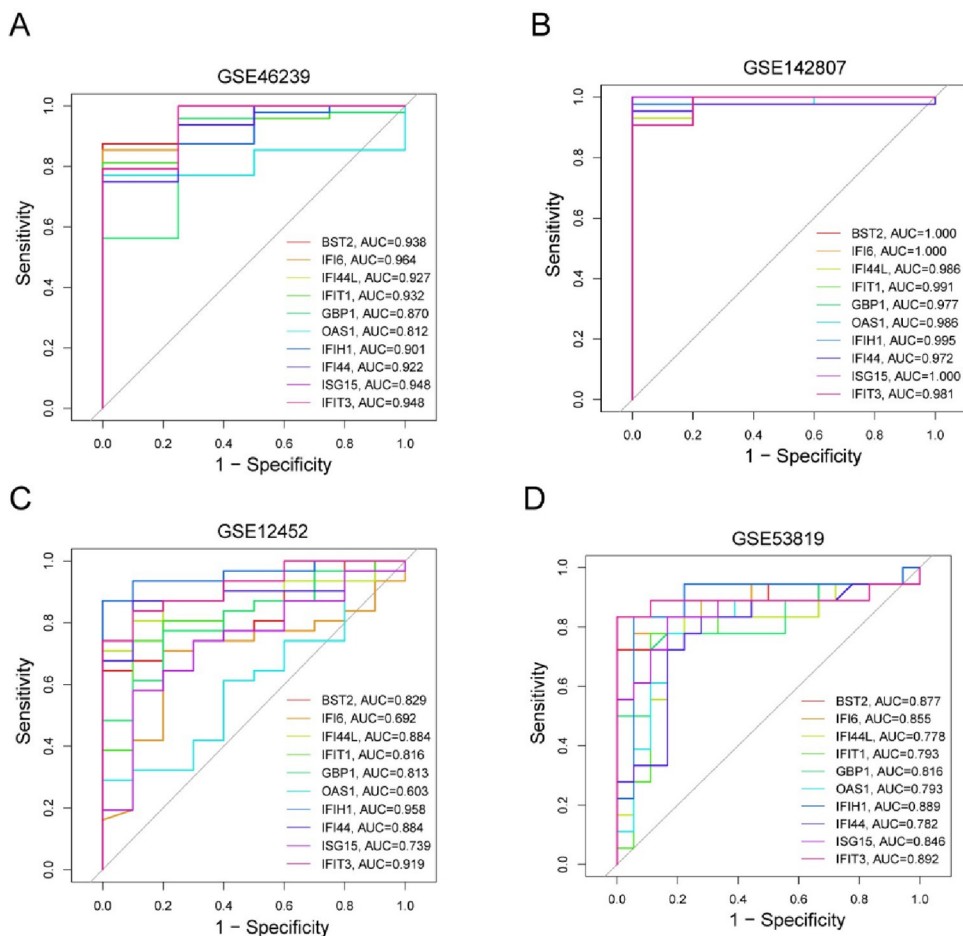

**Fig 9. The evaluation of the independent predictive performance of hub genes in DM and NPC.** (A, C) ROC curve analysis hub genes in the training set, GSE46239 and GSE12452. (B, D) ROC curve analysis hub genes in the validation set, GSE142807 and GSE53819. AUC, area under the curve.

[22]. However, persistent viral infection remains controversial [23]. Coronavirus-19 mimics the symptoms of DM and causes myositis [24]. As DM occurs more frequently in spring, respiratory viral infections may be associated with an increased risk of anti-MDA5-positive DM [25]. Epstein-Barr virus (EBV) infection can trigger a cascade of inflammatory reactions, aggravating autoimmune diseases. In patients with juvenile dermatomyositis, Epstein-Barr nuclear antigen-IgG as well as EBV capsid antigen-IgG were highly positive [26]. EBV proteins can mimic myosin anti-tRNA synthetases, EC-RF4 maps to histamine-tRNA and alanyl-tRNA to BPFL1. Hence, antiviral antibodies may cross-react with tRNA synthetases [27]. Although type I interferon response is predominant in DM, EBV's latent membrane protein 1 (LMP1) can inhibit IFN-stimulating genes (ISGs) expression by blocking interferon regulatory factor 5 (IRF5) and STAT transcription factors. MDA5 triggers IFN-induced antiviral responses, but MDA5 is not able to sense virus infection since the virus inhibits PP1 dephosphorylation [28].

Viral infection also plays a role in the pathophysiology of cancer. EBV infection is significantly associated with nasopharyngeal carcinoma. EBV BALF2 variants account for more than 80% of the overall risk of NPC in southern China [29]. LMP1 promotes dedifferentiation and proliferation of NPC tumor cells via epigenetic regulation. Additionally, LMP1 increases tumor growth by inhibiting PERK enzyme activity in tumor cells [30, 31]. With the help of an

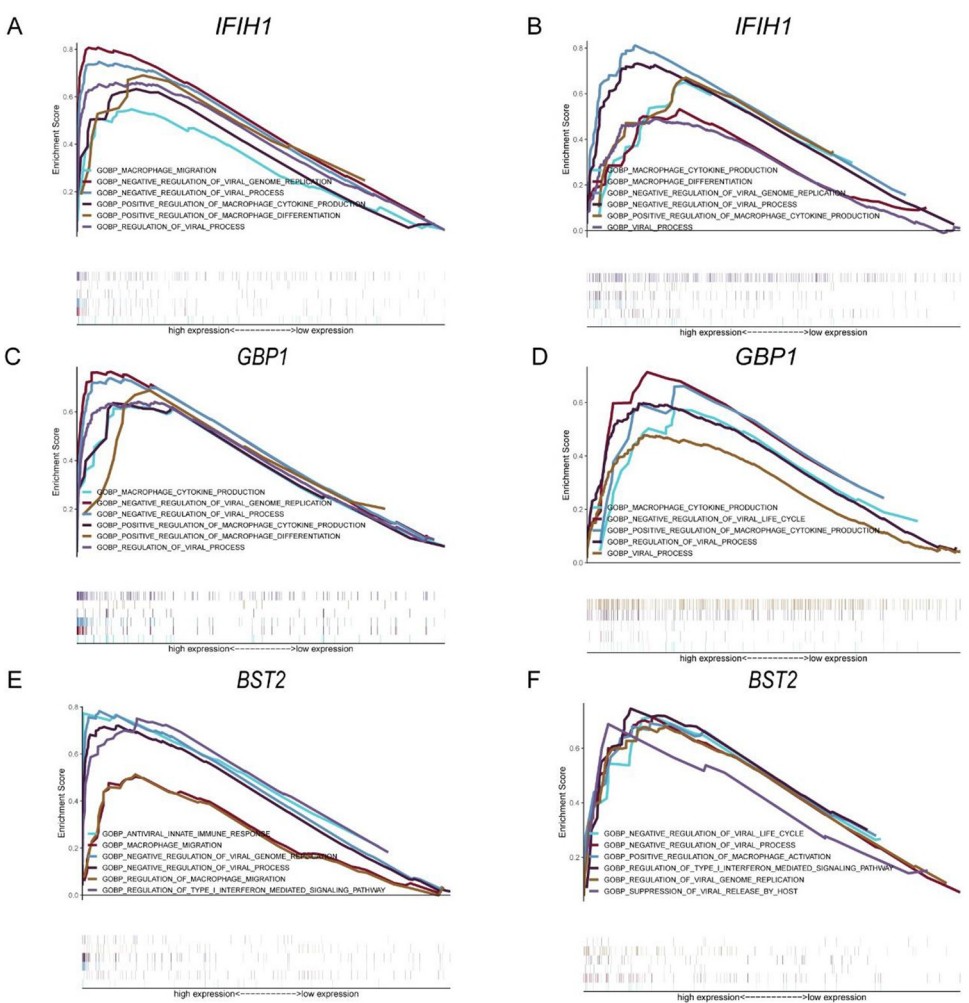

**Fig 10. Gene set enrichment analysis.** (A, B) A merged enrichment plot of IFIH1 from gene set enrichment analysis in GSE142807 and GSE53819, respectively. (C, D) A merged enrichment plot of GBP1 from gene set enrichment analysis in GSE142807 and GSE53819, respectively. (E, F) A merged enrichment plot of BST2 from gene set enrichment analysis in GSE142807 and GSE53819, respectively. (The absolute value of the NES >1 and the nominal p-value<0.05 were considered to be significant).

anti-EBV antibody, NPC can be predicted and screened for early. By releasing DNA from plasma, virus replication is becoming increasingly useful in detection, prediction, and assessment of distant NPC recurrence [32].

DM is prone to malignancy, especially NPC, suggesting some shared pathophysiology. We have identified ten pivotal antiviral genes that are all interferon-stimulated [33]. Down-regulation of antiviral genes will inhibit type I IFN response. Which contribute to virus replication and tumor growth. Viral infections can play a crucial role in DM predisposed to nasopharyngeal cancer, EBV DNA and anti-EBV IgA are significantly higher in DM combined with NPC patients support this point [34].

## The regulation of has-miR-146a on DM-associated NPC

miR-146a is a negative feedback regulator of the natural immune response. In SLE, miR-146a is downregulated, and miR-146a negatively regulates type I interferons expression by

inhibiting the Toll-like receptor signaling pathway [35]. SLE patients' risk variant rs2431697 interacts with the miR-146a promoter to increase miR-146a expression, and miR-146a inhibits type I interferon production from peripheral blood mononuclear cells [36]. In contrast, miR-146a is overexpression in macrophages of RA synovial tissues, and is closely related to disease activity [35]. miR-146a is also a key regulator of Treg cells. miR-146a promotes the maintenance of immune tolerance function of Treg by regulating the expression of target gene STAT1, reducing miR-146a expression in RA Tregs will aggravating RA symptoms [37]. Myositis muscle tissue has been found to overexpress miR-146a which is mainly expressed by leukocytes. It regulates monocyte differentiation and negatively regulates type I interferon [1, 38]. Our study also found that up-regulated 146a negatively affected macrophage M1 function by suppressing antiviral gene expression in DM.

Overexpression of miR-146a promotes viral gene replication following viral infection by targeting TNFR-associated factor 6 (TRAF6) [39]. A knockout or neutralization of miR-146a increased type I interferon production and inhibited viral replication by restoring IRAK1 and TRAF6 expression [40]. Our study found that miR-146a negatively regulated several antiviral genes (*IFIT3*, *ISG15*, *IFI44*, *IFI44L*, *and IFIT1*). while macrophage M1 infiltration was positively correlated with antiviral gene expression. miR-146a can inhibit macrophage M1 infiltration by regulating the expression of macrophage REG3A protein [41]. Inhibiting macrophage M1, an immune checkpoint, may lead to immune evasion by tumor cells [20].

MiR-146a negatively regulates innate immunity and interferon response, thereby promoting viral replication and tumor immune escape. This may promote dermatomyositis complicating nasopharyngeal carcinoma.

## IFIH1

The *IFIH1* gene encodes the MDA5, an important sensor of viral dsRNA. In response to viral RNA stimulation, MDA5 promotes macrophage M1 polarization and is crucial in preventing pathogen infection [42]. In response to virus infection, type I IFN and inflammatory cytokines can activate MDA5, and excessive activation of the type I IFN pathway can lead to autoinflammatory diseases. MDA5+ DM may thus be predisposed by viral infection [43]. Several autoimmune diseases are associated with IFIH1. Increased serum IFN and upregulation of IFIH1 result in persistent disease activity when *IFIH1* mutations are present in SLE patients [44]. *IFIH1* polymorphisms (rs1990760 and rs3747517) appear to be closely related to high antibody titers against dsDNA, Smith proteins, and ribonucleoproteins [45]. In addition, increased expression of MDA5 triggers tumor cell apoptosis and enhances antitumor immunity [46]. Alternatively, *IFIH1* downregulation may contribute to DM complicating NPC.

## GBP1

The guanylate-binding protein1 (GBP1) is GTPases that can be activated by interferons. Its n-terminal structural domain can directly bind to pathogens or recognize pathogen components and activate inflammasomes, thus activating the innate immune system [47]. In rheumatoid autoimmune disease patients characterized by chronic inflammatory vascular disease, GBP1 increases and serves as a biomarker [48]. GBP1 exhibits anti-tumor effects by inhibiting tumor cell proliferation. However, Cancer cells can evade the IFN-γ-based Th1 immune response through downregulated GBP1 [49]. In our study, *GBP1* was down-regulated and may result in a decrease in antiviral activity and easier for tumors to grow.

## IFIT3

*IFIT3* encodes a protein that belongs to the IFN-inducible tetrapeptide repeat-containing sequence (IFIT) family. IFIT3 and IFIT1 interact synergistically to enhance viral RNA recognition and antiviral activity [50]. IFIT3 is strongly upregulated during macrophage M1 polarization [51], indicating its important role in virus defense. Patients with SLE have significantly elevated levels of IFIT3, which can activate the cGAS/STING pathway through its interaction with STING and TANK-binding kinase 1 to produce type I IFN [52]. IFIT3 can enhance the anti-liver cancer effect of IFN-α by binding transcriptional activators STAT1 and STAT2 [53]. However, IFIT3 can also promote tumor progression and metastasis in squamous cell carcinoma of the head and neck [54]. The specific role of IFIT3 in NPC needs to be investigated.

## BST2

BST2 (CD137, tetherin) is an interferon-induced transmembrane protein whose topology allows it to anchor viral particles directly to cells, preventing viral transmission [55]. In NPC, overexpression of BST2 inhibits the endogenous replication of EBV and increases tumor cell resistance to cisplatin, indicators of poor outcomes. Cisplatin-sensitive nasopharyngeal carcinoma cells showed downregulation of BST2 expression and were more susceptible to apoptosis [56]. Lupus patients had low methylation and high expression of interferon regulatory genes, including *BST2*, *IFIT3*, *IFI44L*, and *IFIT1* [57]. In interferon-hypersensitive autoimmune diseases, including SLE and DM, needs to be explored further.

Studies on dermatomyositis complicated by nasopharyngeal carcinoma are relatively scarce. In this study, we discovered that dermatomyositis-associated nasopharyngeal cancer may be a result of viral gene replication and down-regulation of type I interferon response. However, our study also had some limitations. For example, the small sample size in some datasets may lead to selection bias. Additionally, our findings are mainly based on bioinformatics analysis, more clinical trials and in vitro experiments will be needed for further study.

In conclusion, this study identified the pivotal genes that play a role in the development of DM and NPC. Negative regulation of viral gene replication and type I interferon response may be a crucial mechanism for the concurrence of NPC in DM. Additionally, miR-146a's regulation of virus infection and IFN-I may provide new opportunities for the diagnosis and treatment of DM in combination with NPC. Nasopharyngeal carcinoma screening may be necessary in individuals with DM who have persistent viral replication.

## Supporting information

**S1 File.**
(DOCX)

**S1 Table. Generalize of four datasets in DM and NPC.**
(DOCX)

**S2 Table. GSE46239 data.**
(XLS)

**S3 Table. GSE12452 data.**
(XLS)

**S4 Table. GSE142807 data.**
(XLS)

**S5 Table. GSE53819 data.**
(XLS)

**S6 Table. miRNA in DM.**
(XLS)

**S7 Table. miRNA in NPC.**
(XLS)

**S8 Table. miRNA of hub genes.**
(XLS)

## Acknowledgments

We want to thanks GEO, Cytoscape, GSEA, GeneCards, miRWalk, and miRTarbase databases or software for free use.

## Author Contributions

**Conceptualization:** Xiuqin Zhong, Jingjing Shang, Rongwei Zhang.

**Data curation:** Xiuqin Zhong, Jingjing Shang.

**Formal analysis:** Xiuqin Zhong.

**Investigation:** Jingjing Shang, Rongwei Zhang.

**Methodology:** Xiuqin Zhong, Jingjing Shang, Rongwei Zhang.

**Resources:** Xiuqin Zhong, Haitao Niu.

**Software:** Xiuqin Zhong.

**Supervision:** Xiuling Zhang, Le Yu, Haitao Niu.

**Validation:** Xiuling Zhang, Le Yu, Haitao Niu.

**Visualization:** Jingjing Shang.

**Writing – original draft:** Xiuqin Zhong.

**Writing – review & editing:** Haitao Niu, Xinwang Duan.

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
