## [Decision Letter · Decision Letter 0]

6 Sep 2023

PONE-D-23-24175Explore the shared molecular mechanism between dermatomyositis and nasopharyngeal cancer by bioinformatic analysisPLOS ONE

Dear Dr. Duan,

Thank you for submitting your manuscript to PLOS ONE. After careful consideration, we feel that it has merit but does not fully meet PLOS ONE’s publication criteria as it currently stands. Therefore, we invite you to submit a revised version of the manuscript that addresses the points raised during the review process.

We look forward to receiving your revised manuscript.

Kind regards,

Gurudeeban Selvaraj

Academic Editor

PLOS ONE

3. Please amend the manuscript submission data (via Edit Submission) to include author Haoguang Li.

Reviewers' comments:

Reviewer's Responses to Questions

**Comments to the Author**

1. Is the manuscript technically sound, and do the data support the conclusions?

Reviewer #1: Partly

2. Has the statistical analysis been performed appropriately and rigorously? 

Reviewer #1: No

3. Have the authors made all data underlying the findings in their manuscript fully available?

Reviewer #1: No

4. Is the manuscript presented in an intelligible fashion and written in standard English?

Reviewer #1: No

5. Review Comments to the Author

Reviewer #1: The author should address all the questions for furter processing. T he manuscript must describe a technically sound piece of scientific research in introduction part, statistical analysis has not been performed appropriately and rigorously,

6. PLOS authors have the option to publish the peer review history of their article (what does this mean?). If published, this will include your full peer review and any attached files.

Reviewer #1: No

---

## [Author Response · Author response to Decision Letter 0]

3 Nov 2023

Dear reviewers:

 Thank you for your decision and constructive comments on my manuscript. We have carefully considered the suggestion of Reviewer and make some changes. We have tried our best to improve and made some changes in the manuscript which we hope meet with approval. Thank you very much for your attention and time. Look forward to hearing from you.

---

## [Decision Letter · Decision Letter 1]

5 Dec 2023

Explore the shared molecular mechanism between dermatomyositis and nasopharyngeal cancer by bioinformatic analysis

PONE-D-23-24175R1

Dear Dr. Duan,

We’re pleased to inform you that your manuscript has been judged scientifically suitable for publication and will be formally accepted for publication once it meets all outstanding technical requirements.

Kind regards,

Gurudeeban Selvaraj

Academic Editor

PLOS ONE

Reviewers' comments:

Reviewer's Responses to Questions

**Comments to the Author**

1. If the authors have adequately addressed your comments raised in a previous round of review and you feel that this manuscript is now acceptable for publication, you may indicate that here to bypass the “Comments to the Author” section, enter your conflict of interest statement in the “Confidential to Editor” section, and submit your "Accept" recommendation.

Reviewer #1: All comments have been addressed

2. Is the manuscript technically sound, and do the data support the conclusions?

Reviewer #1: Yes

3. Has the statistical analysis been performed appropriately and rigorously? 

Reviewer #1: Yes

4. Have the authors made all data underlying the findings in their manuscript fully available?

Reviewer #1: Yes

5. Is the manuscript presented in an intelligible fashion and written in standard English?

Reviewer #1: Yes

6. Review Comments to the Author

Reviewer #1: The authors have effectively addressed all the queries and concerns raised during the review process. The comprehensive revisions and clarifications provided demonstrate a thorough commitment to improving the manuscript. Based on the significant contributions and adherence to scholarly standards, I recommend the acceptance of this manuscript for publication.The study's discoveries, methodology, and insights serve as a valuable contribution to our understanding of the mechanisms underlying between dermatomyositis and nasopharyngeal cancer.

7. PLOS authors have the option to publish the peer review history of their article (what does this mean?). If published, this will include your full peer review and any attached files.

Reviewer #1: No
